# Edit-R1: Unleashing Reasoning-Based Reinforcement Learning for Image Editing

## Abstract

While Reinforcement Learning from Human Feedback (RLHF) has become a pivotal paradigm for text-to-image generation, its application to image editing remains largely unexplored. A key bottleneck is the lack of a robust general reward model for all editing tasks. Existing edit reward models usually give overall scores without detailed checks, ignoring different instruction requirements and causing biased rewards. To address this, we propose *Edit-R1*, which boosts image editing models with a chain-of-thought (CoT) reasoning reward model (RRM). This Edit-RRM breaks instructions into verifiable principles, evaluates the edited images against each principle, and aggregates fine-grained scores to reduce hallucinations and provide more interpretable criteria. To build such an RRM, we first apply supervised fine-tuning (SFT) as a "cold-start" to generate CoT reward trajectories. Then, we introduce Group Contrastive Preference Optimization (GCPO), a reinforcement learning algorithm that leverages human pairwise preference data to reinforce our pointwise RRM. After building the RRM, we use GRPO to train editing models with this non-differentiable yet powerful reward model. Extensive experiments demonstrate that our Edit-RRM surpasses powerful VLMs such as Seed-1.5-VL and Seed-1.6-VL as an editing-specific reward model, and we observe a clear scaling trend, with performance consistently improving from 3B to 7B parameters. Moreover, Edit-R1 delivers gains to editing models like FLUX.1-kontext, highlighting its effectiveness in enhancing image editing.

## 1 Introduction

With the rapid progress of diffusion models, text-to-image (T2I) generation and image editing have advanced dramatically. In T2I generation, Reinforcement Learning from Human Feedback (RLHF) has become a core step in post-training (Gong et al., 2025a; Gao et al., 2025; Wu et al., 2025a). This is reflected in the rise of reward models such as ImageReward (Xu et al., 2023), HPS (Ma et al., 2025), UnifiedReward (Wang et al., 2025c), and RewardDance (Wu et al., 2025b), as well as optimization methods like DPO (Wallace et al., 2024), REFL (Xu et al., 2023), and GRPO (Xue et al., 2025; Liu et al., 2025a). By contrast, image editing research remains centered on pretraining and supervised fine-tuning (SFT) (Wang et al., 2025b; Deng et al., 2025; Batifol et al., 2025), with little exploration of RLHF.

In this paper, we identify two key challenges for applying RLHF to image editing: (1) *Reasoning-enhanced Reward Models (RRMs)*. Existing edit evaluators (Gong et al., 2025b; Wei et al., 2025) typically rely on CLIP-based or general-purpose Vision Language Models (VLMs) architectures to directly output a single score. However, image editing requires a more nuanced evaluation than T2I generation, covering aspects such as preservation of unedited regions, fidelity to the editing instructions, and overall image quality. The straightforward scoring approach often fails to balance these aspects, resulting in bias or hallucinated feedback (Gunjal et al., 2025). There is a critical need for a thinking reward model that reasons across multiple dimensions and leverages test-time computation to improve fidelity. (2) *RLHF Algorithms Compatible with RRMs*. While RLHF algorithms such as REFL have been applied to editing models (Gong et al., 2025b; Ren et al., 2024), they are fundamentally incompatible with our reasoning reward model. Since the RRM generates an explicit multi-step reasoning trace through discrete token sampling before producing a final score, the process is inherently non-differentiable, rendering REFL-style methods inapplicable. Thus, a key challenge is how to leverage such a powerful RRM to achieve stable improvements in downstream editing models.

To address these challenges, we introduce Edit-R1, a framework that leverages an Edit-RRM to enhance image editing. Edit-R1 begins with cold-start training of a reasoning VLM through Supervised Fine-Tuning (SFT), then refines it with reinforcement learning, and finally applies the improved RRM to strengthen

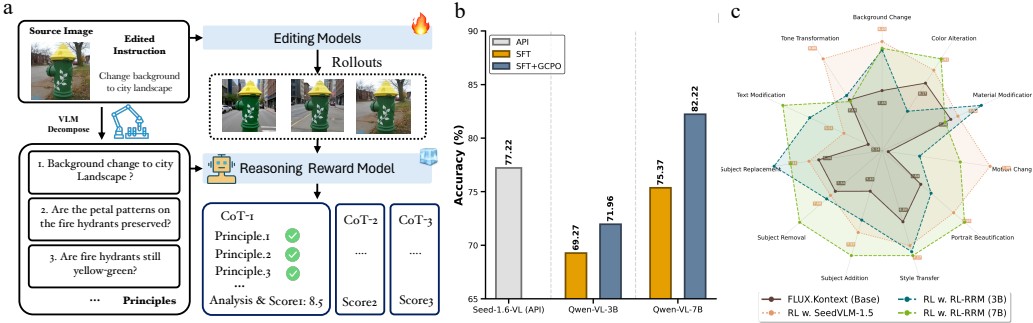

Figure 1: The Edit-R1 Framework: From Reward Modeling to Downstream. (a) RRM Inference Pipeline. The RRM decomposes an instruction into verifiable principles and scores an edited image against them in a single pass. (b) Reward Benchmark Performance. Our final 7B model, trained with SFT and GCPO, reaches 82.22% accuracy, surpassing the Seed-VLM baseline. Each training component contributes to the performance gain. (c) Downstream Application. Using our 7B RL-RRM as a reward signal significantly improves the performance of FLUX.Kontext across multiple editing categories during post-training.

editing models. Our approach is centered on the RRM that evaluates edits based on decomposed principles. Given an instruction and source image, the RRM first decomposes it into verifiable sub-principles, then evaluates the edited image against each principle and aggregates the results into a comprehensive reward. This provides a granular and interpretable feedback signal. The RRM is trained via a two-stage pipeline. (1) The initial "cold-start" stage involves SFT on a diverse dataset of paired (instruction, source/edit image, CoT-scoring trajectory) examples. (2) To enhance the model's reasoning reliability, we then introduce the Group Contrastive Policy Optimization (GCPO), a novel method leveraging human preference data to construct distinct groups of superior and inferior reasoning trajectories. GCPO computes a cross-group reward and intra-group advantage: a cross-group reward for correctly distinguishing between the groups, and an intra-group advantage to capture finer-grained distinctions within them. This GCPO approach, which contrasts entire reasoning pathways, significantly improves the quality of the reward model's inference process and the accuracy of its scoring mechanism. The resulting RRM is then used to finetune the editing model with an Edit-centric GRPO algorithm. Guided by principle-based feedback, the RRM breaks down each instruction, evaluates the generated edit against the derived principles, and produces a holistic reward. This structured signal drives the model to better follow complex user instructions. Our contributions are summarized as follows:

- **Reasoning-enhanced Reward Model.** To our knowledge, we introduce the first Chain-of-Thought (CoT) enabled reward model for image editing. Its fidelity is achieved through a two-stage training process: a "cold-start" phase using SFT on detailed reasoning trajectories, followed by our novel *Group Contrastive Policy Optimization (GCPO)* algorithm. This process significantly enhances both the quality of the RRM's reasoning capability and the accuracy of its final scores.

- **Reinforcement Learning for Image Editing.** We adopt a GRPO-based reinforcement learning algorithm for image editing and show that it effectively leverages the structured, principle-based feedback from our RRM to improve the model's ability to follow complex instructions.

- **Superior Performance and Scaling Trend.** Our 7B Edit-RRM reward model surpasses the evaluative accuracy of powerful proprietary VLMs (e.g., Seed-1.5-VL/Seed-1.6-VL) and exhibits a clear scaling Trend from 3B to 7B. When applied Edit-RRM to open-source editors such as FLUX.1-kontext and Qwen-Image-Edit, our framework achieves substantial gains on challenging editing benchmarks.

## 2 RELATED WORKS

### 2.1 REWARD MODEL FOR GENERATIVE MODELS

Driven by advances in Large Language Models (LLMs), many Reward Models (RMs) are now constructed directly upon them as show in Table 1 (Ren et al., 2024; Wu et al., 2025b; Gong et al., 2025a; Ma et al., 2025). In terms of modeling architecture, two dominant approaches have emerged, including regression-based (Liu et al., 2025b; Wang et al., 2025a) and generative-based (Wu et al., 2025b; Gong et al., 2025b;

| Method | Task | Modeling Paradigm | Point-wise | Reasoning Ability | | |
|---|---|---|---|---|---|---|
| | | | | Use principles | With thinks | learned via RL |
| ImageReward (Xu et al., 2023) | Visual: T2I | Regressive | ✓ | × | × | × |
| VideoAlign (Liu et al., 2025b) | Visual: T2I | Regressive | ✓ | × | × | × |
| WorldPM (Wang et al., 2025a) | Understanding | Regressive | ✓ | × | × | × |
| DeepSeek-GRM (Liu et al., 2025d) | Understanding | Generative | × | ✓ | ✓ | ✓ |
| Pairwise RM (Xu et al., 2025) | Understanding | Generative | × | × | ✓ | ✓ |
| UnifiedReward (Wang et al., 2025c;d) | Multimodal | Generative | × | ✓ | ✓ | ✓ |
| RewardDance (Wu et al., 2025b) | Visual: T2I | Generative | × | × | ✓ | × |
| OneReward (Gong et al., 2025b) | Visual: Edit | Generative | × | × | × | × |
| VisualQuality-R1 (Wu et al., 2025c) | Visual: T2I | Generative | ✓ | × | ✓ | ✓ |
| Skywork-EditReward (Wei et al., 2025) | Visual: Edit | Generative | ✓ | × | ✓ | × |
| **Edit-RRM (Ours)** | **Visual: Edit** | **Generative** | ✓ | ✓ | ✓ | ✓ |

Table 1: Comparison of reward models, highlighting reasoning capabilities. We categorize methods by their foundational characteristics (Task, Modeling Paradigm, etc.) and their support for advanced **Reasoning Ability** components: explicit use of principles, Chain-of-Thought ("thinks"), and reinforcement learning. A checkmark (✓) denotes support. **Edit-RRM (Ours)** is unique in integrating all three reasoning-enhancing features within a generative, point-wise framework for visual tasks.

Hong et al., 2025). The regression-based methods add a regression head for scoring, while the generative methods leverage the model's own generative abilities for assessment and are generally considered more effective at harnessing the base model's power. Regarding input format, methods are either pointwise (Wu et al., 2025c; Wei et al., 2025) or pairwise (Wang et al., 2025c; Wu et al., 2025b). Pointwise methods score a single response independently, while pairwise methods compare two responses to determine a preference. A significant drawback of pairwise approaches is their inability to provide an absolute quality score for a single response, making them ill-suited for direct quality assessment or filtering. In addition, to enhance performance and interpretability, recent work has begun integrating Chain-of-Thought (CoT) reasoning and explicit principles (Liu et al., 2025d; Wang et al., 2025c; Wu et al., 2025b). Building upon these advancements, our work is the first to introduce a generative, pointwise reward model for image editing. Moreover, we employ a CoT process that first decomposes the evaluation into predefined criteria and then aggregates the individual assessments. The model undergoes a two-stage training pipeline, cleverly combining a rationale-based cold-start phase with subsequent reinforcement learning optimization.

## 2.2 REINFORCEMENT LEARNING IN IMAGE EDITING

Recent advancements in Reinforcement Learning from Human Feedback (RLHF) algorithms have demonstrated remarkable efficacy in aligning models with human preferences in the domain of image edting. DreamFuse (Huang et al., 2025) adopts Direct Preference Optimization (DPO) (Wallace et al., 2024) as their optimization method. However, DPO's direct optimization on a preference dataset inherently restricts policy exploration, risking suboptimal convergence. While methods (Gong et al., 2025b; Ren et al., 2024) utilize REFL (Xu et al., 2023) for preference alignment, REFL is often prone to severe reward hacking and requires the reward model to be differentiable. Inspired by the notable success of DeepSeek-R1 (Guo et al., 2025a), many recent works (Xue et al., 2025; Liu et al., 2025a) are now exploring the application of GRPO within the domain of visual generation. A key factor in DeepSeek-R1's success was its reinforcement learning framework with verifiable rewards, which ensured robust training and mitigated reward hacking. Yet, defining such rewards for visual generation remains challenging. To address this, we extend the visual GRPO algorithm with a reasoning-based reward model for image editing, offering structural and principle-driven feedback.

## 3 METHOD

Our method, Edit-R1, comprises a *principle-driven, Reasoning* reward model (RRM) trained with two stages and a reinforcement learning algorithm that leverages this RM to optimize editing models. As detailed in Fig. 2, the training of RRM is a two-stage process. Stage 1 (Cold-start SFT) constructs a large-scale, editing-specific SFT set by decomposing each instruction into keep/follow/quality principles and selecting reliable "think+score" COTs via point-wise verification. Stage 2 (GCPO) trains the RRM with human preference pairs using our novel Group Constative Preference Optimization algorithm and win/loss-ratio rewards to sharpen calibration and mitigate hallucinations, improving the accuracy of the proposed reward model. Finally, we integrate this point-wise reward reasoning model with the GRPO algorithm to elevate the performance of diverse editing models across multiple dimensions.

## 3.1 REWARD MODEL

### 3.1.1 REASONING COT-BASED REWARD MODEL WITH COLD-START

We construct our initial dataset by curating 200K samples from the Imgedit benchmark (Ye et al., 2025). Each sample consists of a reference image $x_{\text{ref}}$ and a corresponding edit instruction $q$. To create a dataset with varied difficulty, we partition it into two equal subsets: i) Random Subset: The first 100K samples are randomly selected from the Imgedit benchmark to represent a general distribution of edits. ii) Hard Subset: The second 100K samples are specifically curated for higher complexity. To achieve this, we employ GPT-4o to filter the remaining data and select for more challenging edit instructions. As illustrated in the top panel of Figure 1, the objective of this stage is to build a supervised dataset for training our reasoning-enabled reward model, following the steps detailed below.

**Step 1: Decomposing Instructions into Principles.** For each reference image and its corresponding edit instruction, we employ the Seed-1.5-VL API to decompose the task into a concise set of verifiable principles using the system prompt in Appendix B.1. These principles span three core aspects of image editing: (a) Keep: elements that should remain unchanged; (b) Follow: modifications required to align with the instruction; (c) Quality: maintenance of generic visual integrity and fidelity. This sample-wise decomposition effectively factorizes the editing task, structuring the model's reasoning process to distinguish between what to preserve and what to modify based on the specific input. Formally, we denote the principle set as $\mathcal{P} = \{p_k\}_{k=1}^{K}$ for each (reference image, instruction) pair. A concrete example of the decomposition is provided in Appendix C.

**Step 2: Large-Scale Quadruple Generation.** For each reference image and corresponding edit instruction, a diverse set of edited candidates is generated using multiple image-editing models, such as Flux-Kontext (Labs, 2024), Bagel (Deng et al., 2025), and SeedEdit3.0 (Wang et al., 2025b). Each candidate $x_{\text{edit}}$, together with the reference input image $x_{\text{edit}}$, the instruction $q$, and the principle set $\mathcal{P}$, forms a *quadruple* $(x_{\text{edit}}, x_{\text{ref}}, q, \mathcal{P})$. This process yields a total dataset of approximately 2 million quadruples.

**Step 3: VLM Reasoning and Point-wise Scoring.** Each quadruple is processed by Vision-Language Models (VLM Pools) employing Chain-of-Thought (CoT) prompting. The VLM first performs a point-wise verification, assessing the edited image against each principle in $P$. Subsequently, it generates a final scalar score representing the overall quality of the edited image, using the system prompt in Appendix B.2. This score is computed as a weighted aggregate of the principle-wise verification outcomes. For more calculation methods and details, please refer to the appendix. To enhance dataset diversity, we sample multiple thinking CoTs for each quadruple by varying system prompts, sampling temperatures, and VLM variants (e.g., Seed-1.5VL-1.5/Seed-1.6-VL), thereby producing multiple "Think + Score" candidates. To be noticed, we will require the VLMs to generate the reasoning trace in a certain format, while verifying principles in JSON format and output the score under "<score> <>", demo shown in Appendix C

**Step 4: External Verification and SFT Data Selection.** All "Think + Score" candidates corresponding to the same quadruple are subjected to an external verification process. This is performed by SeedVLM-1.5, which functions as a *point-wise verifier*. The verifier re-evaluates each principle in $P$ for every reasoning trace and calculates a verification accuracy via the system prompt in Appendix B.3. We then select the thinking CoT that achieves the highest accuracy. The resulting data, comprising the instruction, images, principles, CoT reasoning trace, and final score, constitute the initial Supervised Fine-Tuning (SFT) dataset for the reward model's cold start.

### 3.1.2 REASONING-REINFORCED REWARD LEARNING

Although the reward model possesses effective Chain-of-Thought (CoT) reasoning capabilities following the cold-start SFT phase, we observe that its judgments can be fallible. The model may exhibit hallucinations or struggle to accurately assess the magnitude of edits, such as incorrectly verifying a principle "move to the left of the figure" as successful, but the object has only slightly moved.

To address these limitations, we introduce a Reasoning-Reinforced Reward Learning phase designed to further align the reward model with human preferences. As illustrated in See Fig. 2, this phase employs an inter-reward, intra-advantage based GRPO algorithm, which we term GCPO (Group Constative Preference Optimization), to refine the reward model using human-annotated preference pairs. In this framework, the reward

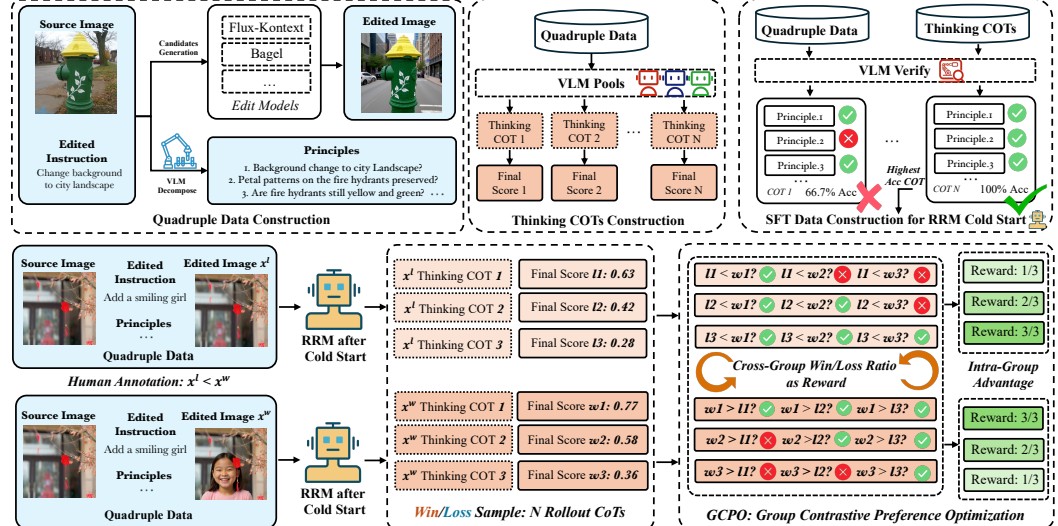

Figure 2: The Training pipeline of Edit-R1 Reasoning Reward Model (RRM). **Top (Cold-Start SFT):** Given an edit instruction and a source image, we generate large-scale quadruple data (instruction, source image, principles, edited image) and employ VLM pools to generate numerous reasoning traces and use another VLM to select the thinking COT with the highest accuracy to build SFT data and cold-start the Reasoning Reward Model (RRM). **Bottom (GCPO):** For each human-labeled preference pair, the reward model generates $N$ thinking-score candidates per image. We compute a win/loss ratio reward by pairwise comparing every candidate in the preferred group against all candidates in the non-preferred group. The win ratio of a preferred candidate equals the fraction of comparisons where its score is *higher* than the opposite group's scores; The loss ratio of a non-preferred candidate equals the fraction where its score is *lower* than the preferred group's scores. The advantage is computed within each preferred or non-preferred group.

model $R_\phi$ itself serves as the policy being optimized, where $\phi$ are its parameters. The "actions" consist of the generated reasoning trace and the final score, which are produced conditioned on the input quadruple data.

**Preference Data.** For this phase, we construct a preference dataset, $\mathcal{D}$, through human annotation. Each sample in this dataset consists of a conditioning context $c = (x_{\text{ref}}, q)$, a verifiable principle $\mathcal{P}$, and a human-labeled preference pair $(x^w, x^l)$. Here $x^w$ denotes the preferred edited image (winner) and $x^l$ is the non-preferred one (loser).

**Win/Loss Ratio Rewards.** We employ pairwise *win/loss ratio* rewards derived from cross-group preference comparisons. For each preference pair $(x^w, x^l)$, the reward model $\mathbb{R}_\phi$ stochastically generates $N$ distinct reasoning traces and their corresponding scores $\{\tau_j^w\}_{j=1}^N$ and $\{\tau_j^l\}_{j=1}^N$ for each image:

$$\tau_j^w = \Phi\big(\mathbb{R}_\phi(x_j^w, c, \mathcal{P})\big), \quad \tau_j^l = \Phi\big(\mathbb{R}_\phi(x_j^l, c, \mathcal{P})\big),$$

where $\Phi(\cdot)$ is an operator that extracts the scalar score from the text output of $\mathbb{R}_\phi(\cdot, \cdot, \cdot)$ via rule-based parsing.

The per-sample win/loss ratios are then defined based on exhaustive pairwise comparisons between the two sets of scores, ignoring ties. The **win ratio** for a preferred candidate $\tau_g^j$ is the fraction of non-preferred candidates it scores *higher* than. Symmetrically, the **loss ratio** for a non-preferred candidate $\tau_b^j$ is the fraction of preferred candidates that score *lower* than it:

$$r_j^w = \frac{1}{N}\sum_{k=1}^N \mathbb{1}\big\{\tau_j^w > \tau_k^l\big\}, \quad r_j^l = \frac{1}{N}\sum_{k=1}^N \mathbb{1}\big\{\tau_j^l < \tau_k^w\big\}, \tag{1}$$

where $N$ denotes the number of reasoning traces generated, $\mathbb{1}\{\cdot\}$ is the indicator function.

**Optimization with GCPO.** After computing the win/loss ratio rewards $\{r_j^w\}_{j=1}^N$ and $\{r_j^l\}_{j=1}^N$ from cross-group comparisons, the original pairing between samples is disregarded for the optimization step. Instead, advantages are computed independently *within* each rollout group (preferred or non-preferred).

Although the rewards originate from paired comparisons, the loss is calculated by partitioning the rollouts into two distinct sets. The advantages are computed as follows:

$$\bar{r}^w = \frac{1}{N}\sum_{j=1}^{N}r_j^w, \qquad \bar{r}^l = \frac{1}{N}\sum_{j=1}^{N}r_j^w, \qquad A_j^w = r_j^w - \bar{r}^w, \qquad A_j^l = r_j^l - \bar{r}^l. \tag{2}$$

Let $r_{t,j}^w(\phi)$ and $r_{t,j}^l(\phi)$ denote the per-token likelihood ratios for the $j$-th rollout and $t$-th token in the preferred and non-preferred groups, respectively. The objective function is the sum of the two groups' clipped surrogate losses, omitting the KL divergence term:

$$\mathcal{L}_{\text{GCPO}}(\phi) = \mathbb{E}_{(c,\mathcal{P},x^w,x^l)\sim\mathcal{D}}\left[\frac{1}{2N}\left(\sum_{j=1}^{N}\frac{1}{T}\sum_{t=0}^{T-1}\min\left(r_{t,j}^w(\phi)A_j^w,\text{clip}\left(r_{t,j}^w(\phi),1-\epsilon,1+\epsilon\right)A_j^w\right)\right.\right.$$
$$\left.\left.+\sum_{j=1}^{N}\frac{1}{T}\sum_{t=0}^{T-1}\min\left(r_{t,j}^l(\phi)A_j^l,\text{clip}\left(r_{t,j}^l(\phi),1-\epsilon,1+\epsilon\right)A_j^l\right)\right)\right]. \tag{3}$$

### 3.2 Reinforcement Learning for Image Editing

We employ the GRPO algorithm (Liu et al., 2025a) and leverages our reasoning-reinforced reward model to provide fine-grained feedback to optimize the image editing model. The editing model, acting as the policy $\pi_\theta(\cdot,c)$ for each sampling step. Following the Group Relative Policy Optimization (GRPO) paradigm, optimizing is as follows: For each conditioning context $c$ sampled from our dataset $\mathcal{D}$, the flow-based editing model $\pi_\theta(\cdot,c)$ generates a group of $G$ edited images $\{x_0^i\}_{i=1}^G$ along with their corresponding generation trajectories $\{(x_T^i,...,x_0^i)\}_{i=1}^G$, where $\{x_T^i\}_{i=1}^G$ is sampled from gaussian distribution.

Our verifiable reward model, $\mathbb{R}_\phi(\cdot,\cdot,\cdot)$ verifies and evaluates each generated image $x_0^i$ based on the context $c$ and the corresponding principle $\mathcal{P}$. Within each group, the advantage $A_i$ for the $i$-th image is calculated by normalizing its reward against the mean and the standard deviation within a group:

$$A_i = \frac{\tau_i - \text{mean}\left(\{\tau_i\}_{i=1}^G\right)}{\text{std}\left(\{\tau_i\}_{i=1}^G\right)+\epsilon_{\text{std}}}, \quad \tau_i = \Phi\left(\mathbb{R}_\phi(x_0^i,c,\mathcal{P})\right), \tag{4}$$

where $\tau_i$ is the holistic reward score provided by our reasoning-based reward model for the $i$-th sample, and $\epsilon_{\text{std}}$ is a small constant for numerical stability. The GRPO training objective is to maximize the expected advantage, incorporating a clipped objective function to prevent excessively large policy updates and a KL-divergence penalty term to regularize the policy $\pi_\theta(\cdot,c)$ and keep it from deviating too far from a reference policy $\pi_{\text{ref}}(\cdot,c)$. This Edit-aware GRPO framework enables the editing model to directly optimize for human-perceived quality and instruction fidelity as captured by our verifiable reward model.

## 4 Experiments

### 4.1 Experimental setups

**Benchmark and Metrics.** For reward model evaluation, we curated a high-quality and diverse set of 5,000 reference images and instructions from ImageEdit (Ye et al., 2025). we then utilized various models, including SeedEdit-3.0 (Wang et al., 2025b), BAGEL (Deng et al., 2025), and FLUX.Kontext (Batifol et al., 2025), to produce several edited outputs for each input. Finally, these generated outputs were manually annotated using pairwise preference comparisons. The accuracy of the reward model in predicting these human-annotated preferences is used as our evaluation metric.

For image editing model evaluation, we adopt GEdit-Bench-EN (Liu et al., 2025c), a standardized benchmark with multi-dimensional automatic metrics. Following the original protocol, we report scores across three key aspects, each assessed by GPT-4.1: semantic consistency (SC), which measures how well the edited image aligns with the given instruction; perceptual quality (PQ), which captures the visual fidelity of the edited image; and overall score (O), computed as the geometric mean of SC and PQ. In addition, we report SC scores for different categories, as presented in Tab. 3.

Table 2: Results on Seed-VL and Qwen-based RRMs. We report the accuracy (%) on our evaluation benchmark, where "—" indicates unsupported configurations. Our Qwen-VL-2.5-7B model trained with Cold-Start and GCPO achieves the best accuracy, surpassing the closed-source reasoning APIs. Note that SFT(T), SFT(V), and SFT(T+V) denote SFT with Think, Verify, and Think+Verify configurations, respectively.

| API | Think | Verify | Think+Verify | |
| --- | --- | --- | --- | --- |
| Seed-1.5-VL (Guo et al., 2025b) | 72.23% | — | 79.28% | |
| Seed-1.6-VL (Guo et al., 2025b) | 71.18% | 69.39% | 77.22% | |
| Model | SFT(T) | SFT(V) | SFT(T+V) | SFT(T+V)+GCPO |
| Qwen-VL-2.5-3B (Bai et al., 2025) | 64.12% | 66.10% | 69.27% | 71.96% |
| Qwen-VL-2.5-7B (Bai et al., 2025) | 68.88% | 70.86% | 75.37% | **82.22**% |

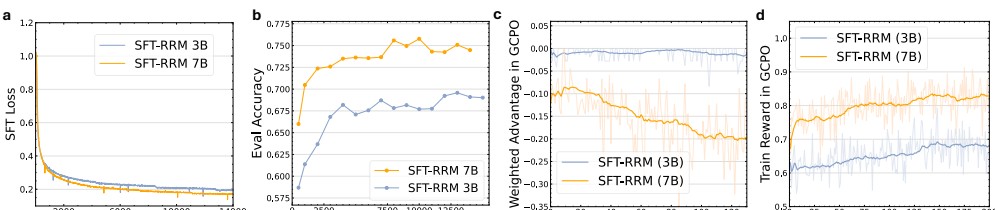

Figure 3: Training dynamics of RRMs. **a**, SFT Loss, showing model convergence and scalability. **b**, SFT evaluation accuracy for the RRMs, showing steady improvement. **c**, Weighted advantage during GCPO training. The weighted advantage is defined as $\frac{1}{G}\sum_{i=1}^{G}\frac{A_i}{L_i}$, with $L$ represents the length of reasoning tokens. The negative value indicates it learns to generate longer reasoning traces for correct judgments. **d**, Training reward during the GCPO phase, showing stable improvement and scalability.

**Implementation Details.** Our reward model (RRM in Edit-R1) is built on the open-source Qwen-VL-2.5 (Bai et al., 2025), with both 3B and 7B parameter sizes. In the cold-start SFT phase in Sec. 3.1.1, we constructed editing pairs using a mixture of models, including SeedEdit-3.0 (Wang et al., 2025b), BAGEL (Deng et al., 2025), and FLUX.Kontext (Batifol et al., 2025). For the GCPO phase in Sec. 3.1.2, we further collected 10k human-annotated preference pairs.

For editing model optimization, we apply our Edit-R1 framework to two strong open-source models: FLUX.Kontext (Batifol et al., 2025) and Qwen-Image-Edit (Wu et al., 2025a). The models are optimized using the GRPO strategy described in Sec. 3.2, with our trained RRM serving as the reward signal. We adopt Flow-GRPO (Liu et al., 2025a) with a group size of $G=24$ and a KL penalty coefficient of $\beta=0.04$.

## 4.2 REWARD MODEL PERFORMANCE

Our proposed Edit-R1 framework yields a state-of-the-art reward model for predicting human preferences. As shown in Tab. 2, Our 7B Edit-RRM, trained via a two-stage pipeline combining Supervised Fine-Tuning (SFT) and GCPO, achieves an accuracy of 82.22%, surpassing the strong Seed-1.5-VL baseline (79.28%). This result highlights the effectiveness of our two-stage training strategy.

**Data Composition.** The foundation of our model's performance is a meticulously curated SFT dataset. Ablation studies in Tab. 2 reveal that both the "Think" (reasoning generation) and "Verify" (filtering) components are critical for creating the Chain-of-Thought (CoT) SFT data. Specifically, removing the "Verify" step results in a significant accuracy degradation for the 7B model (from 75.37% to 70.86%), highlighting the importance of rigorous filtering. Meanwhile, the reasoning traces from the "Think" step provide essential supervisory signals that enhance the model's evaluative capabilities. These findings demonstrate that "Think" and "Verify" are crucial for constructing high-quality SFT data and achieving superior reward modeling performance.

**GCPO Training.** Following the SFT stage, we employ GCPO to further refine the reward model's reasoning capabilities. This reinforcement learning phase yields substantial accuracy improvements for both model scales. As detailed in Tab.2, the 3B model's accuracy rises from 69.27% to 71.96%, while the 7B model improves from 75.37% to 82.22%, thereby outperforming the Seed-1.5-VL baseline. Analysis of

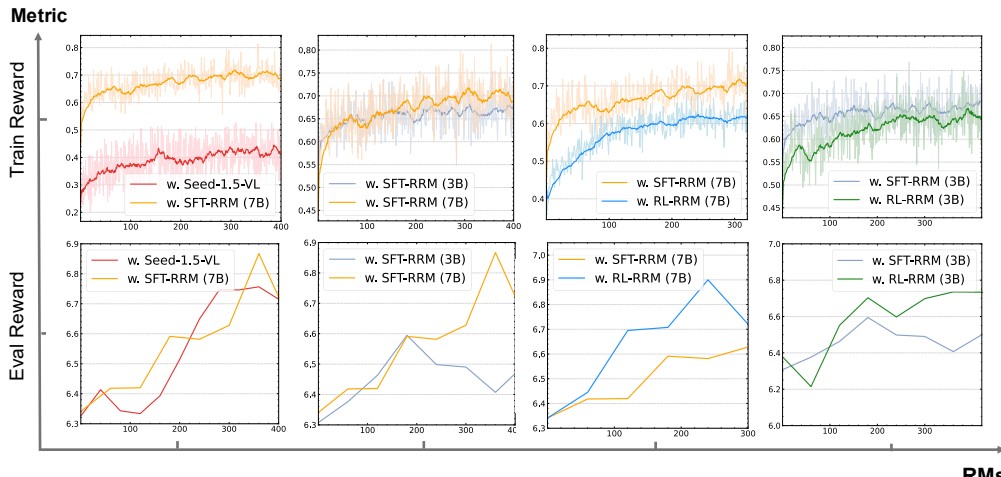

Figure 4: Training dynamics of editing model optimization with different RRMs. The first row shows the training reward, and the second row shows the evaluation reward. Here, SFT-RRM denotes a reward model trained without GCPO, while RL-RRM denotes its counterpart trained with GCPO. First column: our SFT-RRM (7B) produces a reward signal that is as stable and effective as the Seed-1.5-VL. Second column: the SFT-RRM 7B exhibits stronger scalability, providing more reliable supervision and yielding better performance than the SFT-RRM 3B. Third and fourth columns: refining the RRM with GCPO results in consistently higher evaluation rewards, indicating that the RRM trained with GCPO acts as a stricter and more robust evaluator.

the training dynamics (Fig.3c,d) reveals a qualitative shift: the 7B model learns to generate more elaborate reasoning for correct judgements. This indicates that GCPO enhances the model's intrinsic reasoning capabilities, rather than merely overfitting to reward scores.

**Model Parameter Scaling.** Our analysis of model scaling, comparing 3B and 7B variants under identical training configurations, confirms the benefits of increased model capacity. The larger 7B model consistently outperforms its 3B counterpart across all experimental settings, For instance, after the SFT and SFT+GCPO stages, the 7B model achieves accuracies of 75.37% and 82.22% respectively, substantially higher than the 69.27% and 71.96% achieved by the 3B model. This performance gap, corroborated by the 7B model's lower training loss (Fig. 3a), highlights the effectiveness of model scaling in evaluating the edited images.

### 4.3 IMAGE EDITING PERFORMANCE

Fig. 6 illustrates the reward curves of our different RRMs during RL optimization. Our SFT-RRM (7B) delivers reward signals that are as stable as those from the closed-source Seed-VL, while providing higher evaluation rewards. Increasing the mode size from 3B to 7B, the improvement to the editing model is larger, confirming the benefits of scaling. Furthermore, refining RRMs with GCPO leads to higher evaluation rewards than their SFT-only counterparts, showing that GCPO improves RRMs for becoming stricter and more effective evaluators.

**Overall Performance.** As shown in Tab. 3, our framework demonstrates strong performance on both model families. Optimizing FLUX.Kontext with our RL-RRM (7B) boosts its Overall Score (O) from 5.77 to 6.24 and its Semantic Consistency (SC) score from 6.27 to 6.86. This result exceeds the performance under the RL using the closed-source Seed-1.5VL (O: 6.03, SC: 6.74), confirming that our RRM is a highly effective and competitive reward source for policy optimization. The training stability, illustrated in Fig. 6, further validates this, showing that our SFT-RRM (7B) provides a reward signal as stable and reliable as the SeedVLM-1.5. In this paper, we mainly focus on the SC score instead of the Overall score since the scoring of image quality via the VLM isn't robust and reliable. Therefore, we report detailed, category-wise SC scores to provide a more robust measure of editing performance. Meanwhile, we carried out experiments on the SOTA open resource, Qwen-Edit, whose overall score showed a modest improvement from 7.45 to 7.50. This is largely because the baseline model already performs exceptionally well on the GEdit-Bench-EN benchmark, achieving near-perfect results in most categories. However, the validity of Edit-R1 is highlighted in its ability to address the model's specific weaknesses. Notably, as shown in the final row of Tab. 3, our framework yields a significant 15.21% relative gain (from 4.01 to

Table 3: Detailed performance comparison on the GEdit-Bench-EN (Full set). Higher scores are better. **Bold** scores highlight the best result within each model family. Columns 1–11 report SC scores for different editing categories (see Appendix for details).

| Model | Category SC | | | | | | | | | | | Overall | | |
|---|---|---|---|---|---|---|---|---|---|---|---|---|---|---|
| | 1 | 2 | 3 | 4 | 5 | 6 | 7 | 8 | 9 | 10 | 11 | SC ↑ | PQ ↑ | O ↑ |
| **Edited Models** | | | | | | | | | | | | | | |
| Step-Edit (Liu et al., 2025c) | 8.77 | 8.90 | 7.52 | 4.35 | 4.10 | 7.73 | 8.56 | 7.81 | 8.26 | 2.82 | 7.30 | 6.53 | 6.72 | 5.90 |
| UniPic2 (Wei et al., 2025) | 8.07 | 8.70 | 6.75 | 3.57 | 4.78 | 7.13 | 8.36 | 8.36 | 7.87 | 5.39 | 7.97 | 6.84 | 7.24 | 6.41 |
| Bagel (Deng et al., 2025) | 8.54 | 8.32 | 7.42 | 4.97 | 5.07 | 7.71 | 8.75 | 8.03 | 8.22 | 7.14 | 6.62 | 7.32 | 7.02 | 6.65 |
| GPT-4o | | | | | | | | | | | | 7.74 | 8.13 | 7.49 |
| **FLUX.Kontext Family** (Batifol et al., 2025) | | | | | | | | | | | | | | |
| FLUX.Kontext | 7.65 | 8.17 | 6.90 | 3.02 | 3.54 | 6.86 | 7.43 | 7.54 | 6.95 | 5.14 | 7.85 | 6.27 | 7.25 | 5.77 |
| RL w. SeedVLM-1.5 | 8.23 | 8.41 | 7.00 | **5.00** | 4.05 | 7.17 | 7.93 | 7.60 | 7.11 | 5.51 | 8.60 | 6.74 | 6.44 | 6.03 |
| RL w. SFT-RRM (3B) | 7.48 | 8.25 | 6.65 | 3.20 | 4.01 | 7.25 | 7.77 | 7.15 | 7.61 | 5.70 | 7.90 | 6.52 | 6.26 | 5.63 |
| RL w. RL-RRM (3B) | 8.13 | 7.65 | **7.33** | 3.63 | 3.70 | 7.25 | 7.78 | 7.65 | **7.68** | 6.03 | **7.93** | 6.67 | 7.09 | 6.10 |
| RL w. SFT-RRM (7B) | **8.25** | 8.52 | 7.10 | 3.85 | 3.68 | 7.27 | **8.26** | 7.68 | 7.38 | 6.44 | 7.80 | 6.81 | **7.25** | 6.20 |
| RL w. RL-RRM (7B) | 8.15 | **8.62** | 6.82 | 4.42 | **4.22** | **7.30** | 8.21 | **7.99** | 7.41 | **6.44** | 7.82 | **6.86** | 7.20 | **6.24** |
| **Qwen-Edit Family** | | | | | | | | | | | | | | |
| Qwen-Edit | **8.85** | 9.02 | **8.20** | 4.01 | 6.04 | 7.61 | **8.8** | 8.32 | 8.74 | 9.00 | 8.37 | 7.94 | **7.78** | 7.45 |
| RL w. RL-RRM (7B) | 8.75 | **9.05** | 8.10 | **4.62** | **6.17** | **7.76** | 8.72 | **8.43** | **8.79** | **8.69** | 8.32 | **7.99** | 7.76 | **7.50** |

4.62) in the challenging Motion Change category (column 4), demonstrating its effectiveness in enhancing performance even on highly optimized models. We also show the quantitative results in Fig. F. Our in-depth analysis indicates that the underlying improvements primarily involve an enhanced range of motion and understanding relatively abstract concepts, such as age.

**Scalability of Reward Models.** The quality of the editing model scales directly with the strength of the reward model. Comparing our 3B and 7B RRMs, the larger model consistently provides better guidance, particularly in aligning outputs with user instructions. For example, Tab. 3 shows that optimizing with SFT-RRM (7B) raises the SC score from 6.52 to 6.84, yielding stronger instruction adherence. This gain, together with improved perceptual quality, increases the final overall score from 5.63 to 6.20. The training curves in Fig. 6b further illustrate this trend that RRM 7B delivers more robust reward signals.

**Impact of GCPO.** GCPO brings clear advantages when applying RRMs to downstream editing tasks. As shown in Tab. 3, RRM with GCPO consistently outperforms its SFT counterpart, with the 3B case improving from an overall score of 5.63 to 6.10. Training curves in Fig. 6 further reveal the mechanism. Although RL-RRMs provide lower training rewards, they yield higher evaluation rewards, indicating that GCPO transforms the reward model into a stricter and more reliable evaluator. This stricter supervision pushes editing models to adhere more closely to human preferences and achieve higher-quality outputs, confirming the necessity of GCPO for maximizing editing model performance.

**Qualitative Analysis.** Beyond quantitative metrics, the qualitative improvements are equally compelling. As shown in the Appendix (Fig. 8), models optimized with our framework exhibit markedly better instruction adherence and visual fidelity. In challenging edits such as Subject Replace, our method successfully follows the instruction, whereas the baseline fails. For localized edits such as Color Alter, our approach precisely modifies the target object without introducing global color shifts or artifacts. These qualitative results provide concrete evidence of the practical effectiveness of our framework in real-world editing scenarios.

## 5 CONCLUSION

We introduce Edit-R1, a novel framework designed to enhance image editing through Reinforcement Learning from Human Feedback (RLHF). Its core is a Reasoning-Enhanced Reward Model (RRM), trained via a "cold-start" SFT phase and our innovative GCPO algorithm, which achieves evaluation accuracy that surpasses powerful proprietary models. By integrating this powerful RRM with a GRPO-based RL algorithm, our framework substantially enhances the instruction-following capabilities of state-of-the-art editing models like FLUX.Kontext.

## ETHICS STATEMENT

This research aims to develop reward models and preference optimization methods for visual generative models. We are committed to upholding the highest ethical standards throughout our work. Our experiments are conducted using publicly available models (e.g., Qwen-VL), APIs, and datasets. The synthetic data central to our work is generated either from text-based prompts or by leveraging existing public image datasets. We acknowledge that some of these public datasets may contain images of individuals. To mitigate privacy concerns, we have strictly adhered to the licensing terms and usage policies of each dataset.

## REPRODUCIBILITY STATEMENT

We are fully committed to ensuring the reproducibility of our research. To facilitate this, all essential components required to replicate our findings will be made publicly available upon publication. We believe these resources provide the community with all the necessary components to verify our results, build upon our work, and foster further innovation in this area.

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

## A THE USE OF LARGE LANGUAGE MODELS(LLMs)

Large Language Models (LLMs) were only used to correct grammar errors and polish the writing. They were not involved in research ideation, experiment design, analysis, or other substantive contributions.

## B SYSTEM PROMPT

### B.1 SYSTEM PROMPT FOR DECOMPOSING THE PRINCIPLES

To automatically decompose the principles, we designed a detailed system prompt for the large vision-language model. This prompt utilizes a few-shot learning approach, providing the model with several complete examples before presenting it with a new task. The prompt is structured to guide the model's role, define the task requirements, specify the output format, and provide context through examples.

Below are the system prompt used to decompose the principle:

```
You are an expert
    ↪  image editing evaluator. Your task is to generate evaluation points for a new image editing task.
### Reference Example:
Example: Instruction: Convert the original image to anime style
Principles:
[
  {
    "question": "Is the generated image converted to an anime style based on the original image?",
    "category": "Instruction Following"
  },
  {
    "question": "Does the
        ↪  character in the generated image retain the hair and facial features from the original image?",
    "category": "Feature Preservation"
  },
  {
    "question": "Does
        ↪  the character's clothing in the generated image retain the features from the original image?",
    "category": "Feature Preservation"
  },
  {
    "question": "Does the character's pose in the generated image remain consistent with the original image?",
    "category": "Feature Preservation"
  },
  {
    "question": "Do the background
        ↪  elements like the table, sofa, bed, and window retain their original features and layout?",
    "category": "Feature Preservation"
  },
  {
    "question": "Apart from
        ↪  the main background elements mentioned, are other details from the original image preserved?",
    "category": "Feature Preservation"
  },
  {
    "question": "Is the generated image free of significant structural problems?",
    "category": "Image Quality"
  },
  {
    "question": "Is the clarity and overall quality of the generated image good?",
    "category": "Image Quality"
  },
  {
    "question": "Does the scene with the character in the generated image look natural?",
    "category": "Image Quality"
  }
]
### Task Requirements:
Generate 10 evaluation points for the new image editing task, with the following distribution:
1. 3-4 points for "Instruction Following" (to assess the implementation of the edit).
2. 3-4 points for "Feature Preservation" (to assess the retention of original features).
3. 2-3 points for "Image Quality" (to assess the quality of the resulting image).
### Output Format:
A JSON array, where each element contains a 'question' field and a 'category' field.
### New Task:
Instruction: {Edit Instruction}
Image: <image>
Please generate all evaluation points:
```

Listing 1: The detailed system prompt for decomposing the principles given the source image and edit instruction.

### B.2 System Prompt for Reward Model Evaluation

To quantitatively score the edited image, we employ a Reasoning Reward Model (RRM). The RRM is guided by a detailed system prompt designed to act as a professional evaluator. This prompt instructs the model to firstly assess the edited image based on the decomposed principles, considering the edit instruction, and performing a holistic analysis of the output quality. The prompt defines a structured evaluation process, including rule definitions, an execution flow, and a strict output format. The complete system prompt provided to the RRM is detailed below.

### B.3 System Prompt for VLM Verification

Our data annotation pipeline incorporates a VLM-based verification stage to generate high-quality, fine-grained evaluation data. This process is divided into two steps, each guided by a specific system prompt: **Verification** and **Selection**. First, a powerful VLM acts as a **Verifier**. It is presented with the source image, the edited image, the instruction, and a list of evaluation points. Crucially, it also receives "reference intermediate judgments"—a collection of Chain-of-Thought (CoT) reasoning excerpts and per-point predictions from multiple candidate models. The Verifier's task is to critically and objectively assess these materials to produce a "gold standard" 0/1 judgment for each evaluation point, effectively acting as an expert human annotator. Second, another VLM acts as a **Selector**. It receives the newly created gold standard and the raw predictions from all candidate models. Following a strict, deterministic ruleset, it calculates the accuracy for each candidate and selects the best-performing one. This two-step process ensures both the quality of the annotations and the objective selection of the most accurate model output. The prompts for both the Verifier and the Selector are detailed below.

## C Inference result of RRM

Listing 4 is the inference result of our RRM, while Fig. 6 shows the results. Then the RRM will summary the results for each principle and output a final score.

## D Category label in quantitative results

Category 1-11 represent Background Change, Color Alteration, Material Modification, Motion Change, Portrait Beautification, Style Transfer, Subject Addition, Subject Removal, Subject Replacement, Text Modification, Tone Transformation.

## E Qualitative Results for FLUX-Kontext

## F Qualitative Results for Qwen-Edit

```
You are a professional evaluation point analyst and image editing evaluator. Your task is to analyze whether
    ↪ a generated image meets a given set of evaluation points, based on the input image and an edit
    ↪ instruction. You must also use divergent thinking based on these points to holistically evaluate the
    ↪ model's editing performance. Your evaluation should not be based solely on the magnitude of the edit
    ↪ ; instead, you must conduct a comprehensive, side-by-side comparison for each evaluation point. If
    ↪ an evaluation point is not met, you must assess the difficulty and complexity of revising the edited
    ↪ image to meet it. Furthermore, you must consider whether elements not mentioned in the instruction
    ↪ or evaluation points (such as the background or secondary subjects) have undergone unreasonable
    ↪ changes. If they were not supposed to change but did, points should be deducted accordingly.
## Input:
- Original Image: <image>
- Edited Image: <image>
- Edit Instruction: {{EDIT_INSTRUCTION}}
- Evaluation Points: {{EXAM_POINTS}}
## Rule Definition:
- For each evaluation point (e.g., "Was the scene changed from indoors to
    ↪ outdoors?"), you can only assign a score of 0 (not met) or 1 (met). For edits involving a range (e.
    ↪ g., far to near, left to right, male to female, fat to thin), a significant change is required to be
    ↪ considered 'met' unless the magnitude is specified. When considering relative positions, if an object
    ↪ faces the camera, the object's left is considered the right side from the viewer's perspective.
- If you are uncertain about an evaluation point, score it
    ↪ as 0 (not met) and incorporate this uncertainty into your subsequent reasoning for the final score.
- The final score should
    ↪ not be solely dependent on the average of the evaluation point scores. The final score can be any
    ↪ value from 0 to 10, not just integers like 0, 5, 8, or 10. If you are not confident about an integer
    ↪ score, use a decimal. If an evaluation point contradicts the edit instruction (e.g., preserving a
    ↪ watch while the instruction is to lower the hand, which would hide it), this point should be ignored
    ↪ when calculating the final score. The consistency of newly revealed areas due to object movement
    ↪ requires special attention, while focusing on the consistency of originally un-occluded parts.
- A perfect score on the evaluation points does not guarantee a perfect final score. You
    ↪ must assess if the edited image is directly usable, if the edit magnitude is appropriate, and if it
    ↪ meets psychological expectations. Also, consider if unmentioned elements have changed unreasonably.
- Crucially, if the edited image is nearly identical
    ↪ to the original (i.e., no edit was performed), assign a score of 0. If the instruction involves
    ↪ a single edit, that edit is the most critical part of the task; if the similarity is too high, the
    ↪ image requires major correction, so score it 0. If the edited image has white borders, score it 0.
- Preserve class information. For example
    ↪ , consistency should be judged based on 3D integrity of material and structure. Even if the viewing
    ↪ angle changes, if it's the same object, consistency is considered good. Prioritize the consistency
    ↪ of the main subject, then secondary subjects/objects. Penalize minor inconsistencies but not
    ↪ heavily if the main subject's consistency is maintained. However, for removal tasks, the object must
    ↪ be completely removed, so pay close attention to positional information of small objects or subjects.
- When dealing with positional information, you must output bounding box coordinates in your thought process.
- For positional
    ↪ changes (e.g., from left to right), a significant shift is required; a minor move is not sufficient.
- When evaluating human pose, strictly determine left and right based on the person's orientation.
- If an edit instruction has N points and one is not met,
    ↪ the deduction should be based on the cost of re-editing the current image to fix that specific point
    ↪ . Deduct more points for fixes that require more information or have a lower probability of success.
- When determining the final score, consider the completion status of multiple key points
    ↪ in the instruction, with a focus on the core directive. For any unmet point, think about the future
    ↪ editing cost (e.g., needing more conditions, more information, or modifying more pixels). Compare
    ↪ this cost hypothetically with the cost of completing other unmet conditions to judge the deduction.
- When an evaluation point
    ↪ contradicts the edit instruction (e.g., requiring consistent color tone during a style transfer, or
    ↪ preserving details on a limb that is moved out of view), prioritize achieving the edit instruction.
- Also, when thinking about the final score, consider unmentioned aspects like
    ↪ the main subject, secondary subjects, and background. If they were not supposed to change but did, or
    ↪ if they changed but are inconsistent, this is a hallucination and should be penalized more heavily.
- Additionally, check for quality issues. The image should not
    ↪ have white borders (minor deduction). Also, check for over-sharpening, oversaturation, or color cast.
- When reasoning about the final score, re-check the
    ↪ following aspects in order of importance: quantity, action/state, negations/comparatives, composition
    ↪ /form/function, material, position/state (e.g., hanging), composition, main subject, environment.
## Execution Flow:
Please follow these steps strictly and sequentially. Do not skip or omit any step:
1. For each evaluation point provided in the format
    ↪ `[{'question':, 'category': }]`, evaluate and score it based on a comparison of the before/after
    ↪ images and the edit instruction, strictly adhering to the scoring standards in the [Rule Definition].
2. Based on the above, assign a final score to the edited image from
    ↪ 0 to 10. 0 means completely unusable (e.g., severe artifacts, very difficult to fix manually). 5
    ↪ means partially usable (some good aspects but far from ready). 8 means nearly usable (minor artifacts
    ↪ , inconsistencies, or instruction deviations that can be fixed with minor manual intervention).
3. When positional changes
    ↪ are involved, output bounding box coordinates in your thought process to reflect your analysis of
    ↪ the position, and then judge if the edit is valid based on the scale of change defined in the rules.
4. Finally, assess
    ↪ the difference between the before and after images to confirm that an edit has actually occurred.
## Output Format:
Produce the output in the following sequence: scores for each evaluation point, the average
    ↪ score of the evaluation points, and finally, the reasoned final score for the generated image.
`[{'question':, 'score': }, ...], {"average_score": } <score> <\score>`
```

Listing 2: The prompt used for the Reward Model (RRM) to evaluate the quality of edited images. This prompt guides the model to score based on predefined decomposed principles.

```
You are a strict image editing verification inspector. Your input includes: an original image, an
    ↪ edited image, an edit instruction, a list of evaluation points, and "reference intermediate judgments
    ↪ " (which are per-point predictions and brief reasoning summaries from multiple candidate models).
Your task is to objectively verify the edits based solely on the images and text, providing a gold-standard
    ↪ judgment (0 or 1) for whether each evaluation point was met, along with a one-sentence reason.
Note: The reference intermediate judgments are for reference only and must not be
    ↪ copied. If the references contradict the images and text, the images and text are the ground truth.
[Rules]
- Each evaluation point can only be scored 0 (not met) or 1 (met).
- If the required
    ↪ magnitude of a change is not specified, a "significant change" is required to be considered 'met
    ↪ '. (e.g., positional changes of less than 10% of the image dimensions are considered insufficient).
- For position-related points, mention the approximate region
    ↪ or bounding box in the reason. A person's left and right are determined by their facing direction.
- If the original and edited images
    ↪ are nearly identical => evaluation points related to the core edit instruction are judged as 0.
- Issues like white borders, severe sharpening, oversaturation, color
    ↪ cast, or structural artifacts can be considered, but the primary task is the per-point 0/1 judgment.
- 'Remove' tasks require
    ↪ complete removal. Prioritize the consistency of the main subject before considering smaller objects.
- If an evaluation point contradicts the edit instruction, prioritize the edit instruction.
## Input:
- Original Image: <image>
- Edited Image: <image>
- Edit Instruction: {{EDIT_INSTRUCTION}}
- Evaluation Points: {{EXAM_POINTS}}
- CoT-1: {{CoT}}
...
- CoT-N: {{CoT}}
[Output]
- Output ONLY a single JSON object (do not output any text outside the JSON), in the following format:
{
  "gold": [
    {"question": "Evaluation
        ↪ point text 1", "gold": 0 or 1, "reason": "Brief reason (can include approximate bbox)"},
    ...
  ]
}
The length and order of the 'gold' array must match the input list of evaluation points.
```

Listing 3: System prompt for the VLM Verifier. It instructs the model to act as a strict inspector to generate gold-standard annotations.

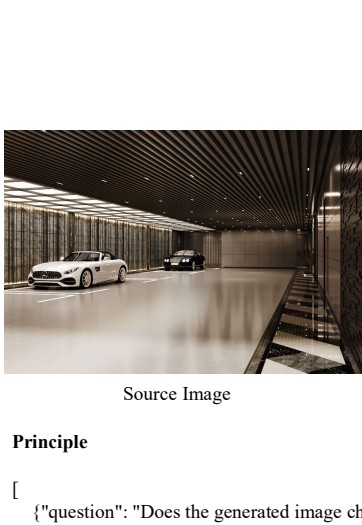 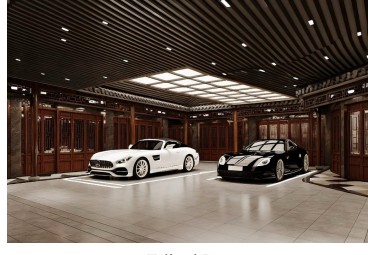

Change the style to Chinese style

Source Image          Instruction                    Edited Image

**Principle**

[
  {"question": "Does the generated image change the garage style from modern to Chinese style?", "category": "Instruction Following"},
  {"question": "Does the generated image contain two sports cars, one white and one black?", "category": "Instruction Following"},
  {"question": "Does the generated image meet the target description of 'two sports cars, one white and one black, parked in a Chinese-style garage'?", "category": "Instruction Following"},
  {"question": "Do the architectural and decorative elements of the garage in the generated image (such as walls, roof, partitions, etc.) reflect the characteristics of Chinese style?", "category": "Feature Preservation"},
  {"question": "Does the overall color tone of the garage in the generated image conform to the characteristics of Chinese style?", "category": "Feature Preservation"},
  {"question": "Is the appearance, shape, and details of the white sports car in the generated image consistent with the original image?", "category": "Feature Preservation"},
  {"question": "Is the appearance, shape, and details of the black sports car in the generated image consistent with the original image?", "category": "Feature Preservation"},
  {"question": "Are there no obvious structural collapses in the generated image (such as abnormal building structures or vehicle proportions)?", "category": "Image Quality"},
  {"question": "Is the clarity and image quality of the generated image good?", "category": "Image Quality"},
  {"question": "Is the integration of Chinese style elements and the vehicles in the generated image natural and harmonious?", "category": "Image Quality"}
]

Figure 5: The Quadratic data demo.

```
The user now needs to analyze the score for each evaluation point. Let's look at them one by one:

First point: ''Does
    ↪  the generated image change the garage style from modern to Chinese style?''. Looking at the original
    ↪  image, it's a modern style (metal, clean lines). After editing, there are wooden doors and windows
    ↪ , and Chinese-style roof decorations (like tiles, wooden frames), so the style has changed. Score: 1.

Second point: ''Does the generated image contain two sports cars, one white and
    ↪ one black?''. The original image has a white and a black car, and the edited image does too. Score: 1.

Third point: ''Does the generated
    ↪  image meet the target description of 'two sports cars, one white and one black, parked in a Chinese-
    ↪ style garage'?''. The style is Chinese, and there are two cars, so it meets the description. Score: 1.

Fourth point: ''Do the architectural and decorative elements of the garage in the generated
    ↪  image (such as walls, roof, partitions, etc.) reflect the characteristics of Chinese style?''.
    ↪  The walls have wooden Chinese-style doors and windows, the roof has Chinese tiles and wooden grilles
    ↪ , and the partitions are also Chinese-style wood. So, it reflects the characteristics. Score: 1.

Fifth point: ''Does the overall color tone of the garage in the generated
    ↪  image conform to the characteristics of Chinese style?''. Chinese style often has warm, woody
    ↪  tones. The edited image has many wooden elements and a warmer tone, which is consistent. Score: 1.

Sixth point: ''Is the appearance,
    ↪  shape, and details of the white sports car in the generated image consistent with the original image
    ↪ ?''. Looking at the white sports car, was the original a convertible? The edited one is a hardtop?
    ↪ The styling details have changed, such as the roof and body lines, so it is not consistent. Score: 0.

Seventh point
    ↪ : ''Is the appearance, shape, and details of the black sports car in the generated image consistent
    ↪  with the original image?''. The original black car was a different model. The edited black car
    ↪ 's styling has changed significantly (front fascia, body lines), so it is not consistent. Score: 0.

Eighth point: ''Are there no obvious structural
    ↪  collapses in the generated image (such as abnormal building structures or vehicle proportions
    ↪ )?''. The building structure and vehicle proportions look normal, with no collapses. Score: 1.

Ninth point: ''Is the clarity and
    ↪  image quality of the generated image good?''. The image is clear and the quality is good. Score: 1.

Tenth point: ''Is the integration of Chinese style elements and
    ↪ the vehicles in the generated image natural and harmonious?''. The combination of Chinese architecture
    ↪  and modern sports cars does not look out of place visually; the integration is natural. Score: 1.

Now, calculate the score for each point:

1. Does the generated image change the garage style from modern to Chinese style?: 1
2. Does the generated image contain two sports cars, one white and one black?: 1
3. Does the generated image meet the target
    ↪  description of 'two sports cars, one white and one black, parked in a Chinese-style garage'?: 1
4. Do the architectural and decorative elements... reflect the characteristics of Chinese style?: 1
5. Does the overall color tone... conform to the characteristics of Chinese style?: 1
6. Is the appearance... of the white sports car... consistent with the original image?: 0
7. Is the appearance... of the black sports car... consistent with the original image?: 0
8. Are there no obvious structural collapses...?: 1
9. Is the clarity and image quality... good?: 1
10. Is the integration of Chinese style elements and the vehicles... natural and harmonious?: 1

Then calculate the average score: (1+1+1+1+1+0+0+1+1+1)/10 = 8/10 = 0.8

Next, consider the final score. The core instruction is to change to a Chinese style. The
    ↪  style change was successful (points 1-5, 8-10 are met), but the vehicle details (points 6, 7) were
    ↪  not preserved. However, the instruction is mainly about changing the style; the vehicles are to be
    ↪ present, not to have their details preserved. Therefore, the core instruction was highly accomplished
    ↪ , and the image quality is good. The points were deducted for vehicle details, but the instruction
    ↪  did not require the vehicle details to be consistent, only that two cars should be present
    ↪ . So the deduction for points 6 and 7 might be due to feature preservation, but the instruction
    ↪  was about style change. So these two points are for feature preservation. Perhaps the user
    ↪ wanted the vehicle appearance to remain unchanged? But the edit instruction is to change the style to
    ↪  Chinese, so is it necessary to preserve vehicle details? Looking at the instruction, it says ''change
    ↪  the style to Chinese style,'' so whether the vehicle appearance is preserved is not the core of the
    ↪  instruction, but the evaluation points include feature preservation. So this needs to be considered.

The total score for all points is 8, and the average
    ↪  score is 0.8. Finally, considering the final score, it can be given a 7. Although the vehicle
    ↪ details have changed, the style change was successful, the image quality is good, and the integration
    ↪  is natural. It is close to usable, with minor issues (vehicle details) that are acceptable.
```

Listing 4: Given the quadruple, the reasoning result of RRM.

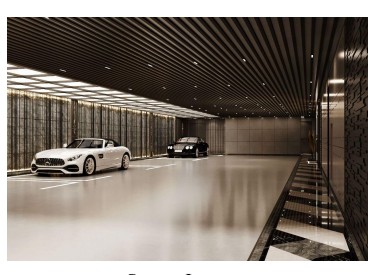   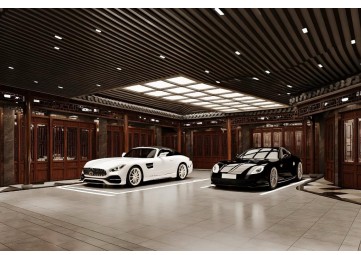

Change the
style to
Chinese style

Source Image                    Instruction                    Edited Image

**Final Output**

[{\"question\": \"Does the generated image change the garage style from modern to Chinese style?\", \"score\": 1},
{\"question\": \"Does the generated image contain two sports cars, one white and one black?\", \"score\": 1}, {\"question\":
\"Does the generated image meet the target description of 'two sports cars, one white and one black, parked in a Chinese-
style garage'?\", \"score\": 1}, {\"question\": \"Do the architectural and decorative elements of the garage in the generated
image (such as walls, roof, partitions, etc.) reflect the characteristics of Chinese style?\", \"score\": 1}, {\"question\": \"Does
the overall color tone of the garage in the generated image conform to the characteristics of Chinese style?\", \"score\": 1},
{\"question\": \"Is the appearance, shape, and details of the white sports car in the generated image consistent with the
original image?\", \"score\": 0}, {\"question\": \"Is the appearance, shape, and details of the black sports car in the generated
image consistent with the original image?\", \"score\": 0}, {\"question\": \"Are there no obvious structural collapses in the
generated image (such as abnormal building structures or vehicle proportions)?\", \"score\": 1}, {\"question\": \"Is the
clarity and image quality of the generated image good?\", \"score\": 1}, {\"question\": \"Is the integration of Chinese style
elements and the vehicles in the generated image natural and harmonious?\", \"score\": 1}], {\"average_score\": 0.8}",
<score>7</score>

Figure 6: The response of RRM after the thinking.

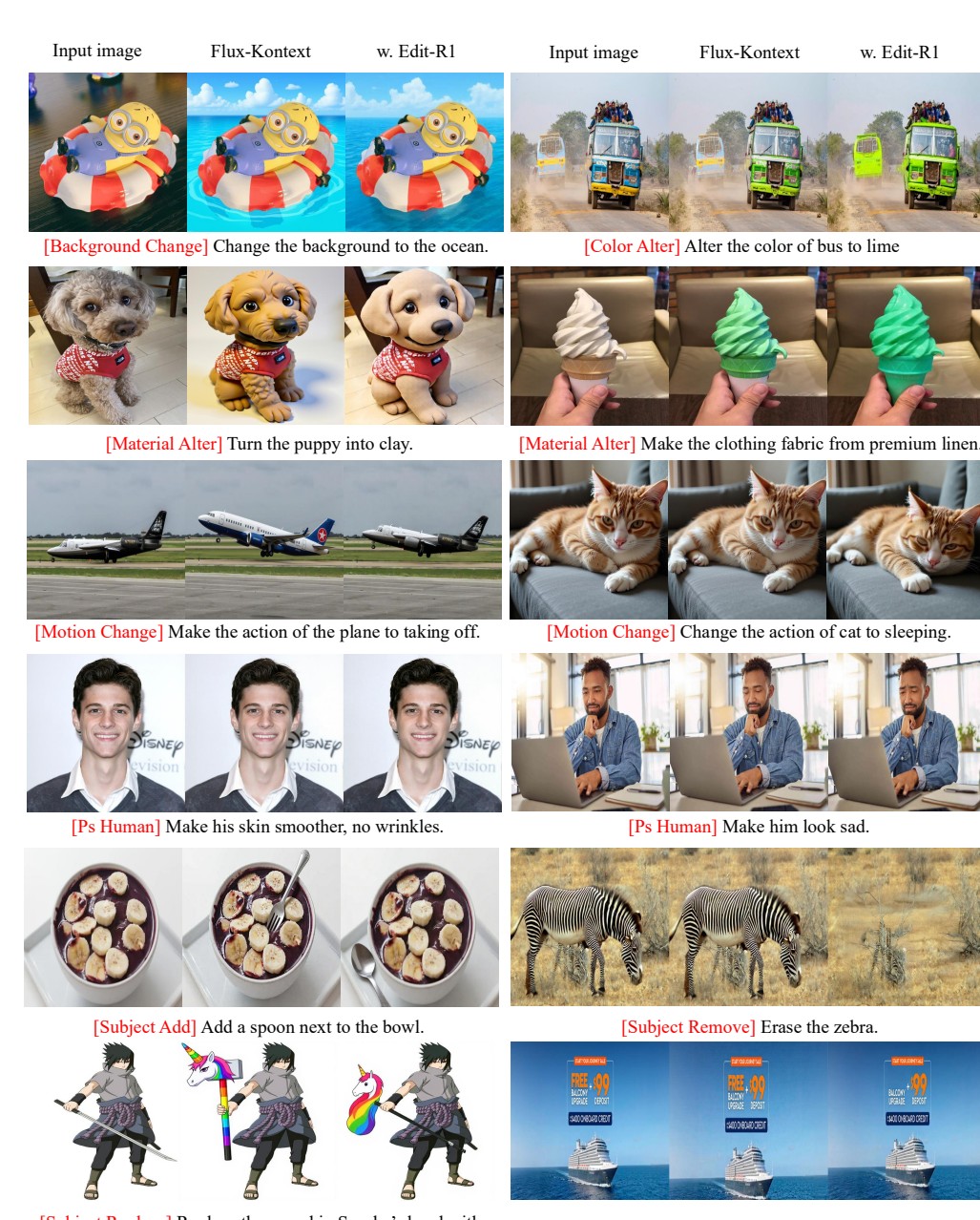

Figure 7: Qualitative comparison of editing results on a diverse set of instructions. For each triplet, we show the input image, the output from the baseline model (FLUX.Kontext), and the output from our enhanced model (FLUX.Kontext w. Edit-R1). Our method demonstrates superior performance improvement for Flux.Kontext across a wide range of editing categories, including tone transformation, color/material alteration, motion changes, human-centric edits, and subject manipulation (add, remove, replace), producing results that better align with user instructions while maintaining high perceptual quality.

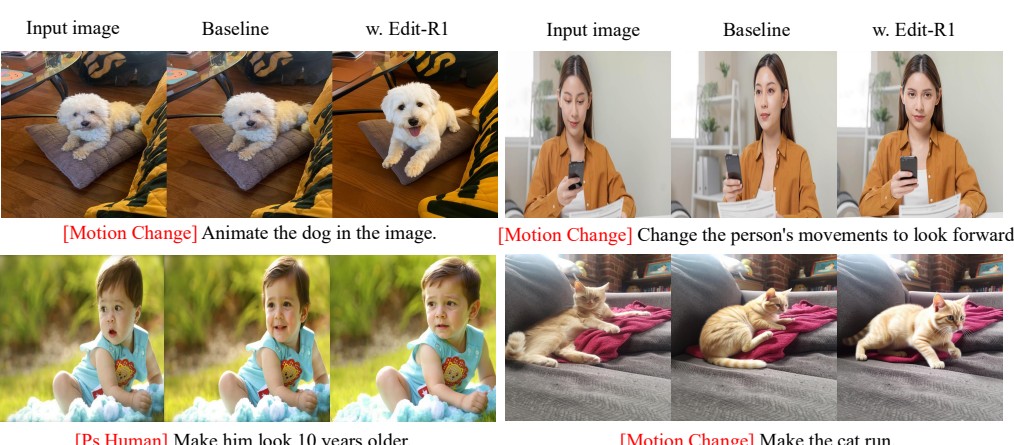

Figure 8: Qualitative comparison of editing results on a diverse set of instructions. For each triplet, we show the input image, the output from the baseline model (Qwen-Edit), and the output from our enhanced model (Qwen-Edit w. Edit-R1). Our method could further improve the performance of Qwen-Edit in motion change, which is a weakness for Qwen-Edit.

