# OpenReview forum: "Edit-R1: Unleashing Reasoning-Based Reinforcement Learning for Image Editing"
_ICLR.cc/2026/Conference — ICLR 2026 Conference Withdrawn Submission_

### Official Review · Reviewer_fGHS · 2025-10-30

**Soundness:** 3
**Presentation:** 2
**Contribution:** 2
**Rating:** 4
**Confidence:** 3

**Summary:**

This paper proposes Edit-R1, a reasoning-based Reward Model (RRM) framework for image editing tasks, aiming to enhance model performance through improved alignment with human preferences. The core idea is to leverage an RRM equipped with Chain-of-Thought (CoT) reasoning, enabling the model to generate not only a score but also the rationale behind its evaluation. This approach aims to more reliably capture human preferences for edited images by providing structured, interpretable feedback.

**Strengths:**

1. Well-motivated design: Requiring the model to generate reasoning traces when scoring enhances the interpretability of the reward signal and improves assessment accuracy and robustness through structured thinking.
2. Comprehensive experiments: The paper conducts thorough experiments on both the performance of the reward model itself and the downstream image editing tasks, providing solid empirical validation.
3. Novel GCPO optimization mechanism: It introduces a GCPO method that computes rewards based on win/loss ratios across groups. This avoids reliance on fixed labels in traditional preference learning and better leverages the uncertainty information from multiple reasoning paths.

**Weaknesses:**

1. Marginal downstream performance gains: When applied to optimize editing models, the overall performance improvement is very small. This raises concerns about whether the high accuracy of the RRM can effectively translate into significant performance gains for the generative model. The authors attribute this to the baseline models already being near-optimal, but they lack validation on more challenging benchmarks or weaker models. Can the authors provide further evidence?
2. High computational cost: The GCPO stage requires generating N reasoning trajectories per image, significantly increasing both training and inference costs. The paper does not provide an efficiency-performance trade-off analysis.
3. Limited model scope for validation: The method is primarily validated on two specific models. Can Edit-R1 be further tested on classic or lightweight editing models to demonstrate broader applicability?
4. Insufficient details on human preference data: While the paper mentions collecting 10k human-labeled preference pairs, it lacks detailed information about the annotation protocol, annotator background, or quality control measures.

**Questions:**

See Weaknesses

---

### Official Review · Reviewer_Vhr4 · 2025-10-31

**Soundness:** 3
**Presentation:** 3
**Contribution:** 3
**Rating:** 4
**Confidence:** 3

**Summary:**

This paper proposes a reinforcement learning method to improve image editing models. They propose to decompose editing instructions to several principles and train a generative reward model to provide both chain-of-thought reasoning and reward scores from different aspects. The reward model is first initialized through supervised fine-tuning and then refined using Group Contrastive Preference Optimization (GCPO) to better align with human preferences. Finally, the trained reward model is used with GRPO to train editing models directly.

**Strengths:**

1. The idea of decomposing editing instructions into principles and evaluating them with CoT reasoning provides a transparent and interpretable approach. It offers more detailed information about the evaluation results and potentially reduces hallucinations of the reward models.

2. To better align the reward model with human preferences, the authors collect a human preference dataset and train the reward model using the GCPO algorithm. Experimental results demonstrate the effectiveness of this reinforcement phase.

3. The paper presents a complete pipeline including data collection, reward model training, and editing model optimization. In the experiments, both the reward model and the downstream image editing performance are evaluated.

**Weaknesses:**

1. The method largely adapts existing RLHF and principle-based reward modeling techniques to image editing domain. Similar methods have been used in other tasks, such as in Deepseek-GRM [1]. Although the paper is content-rich, it feels more like an engineering extension or technical report than a clearly focused research contribution with new conceptual insights.

2. The paper frequently mentions breaking editing instructions into verifiable sub-principles but does not detail how these are objectively verified or how the verification quality is ensured. The process seems to rely on VLM-generated judgments, which are highly subjective and can not be strictly verified.

3. The reward model is trained to evaluate the image editing by several principles, and provide a reasoning process of the scoring criteria. However, this information is not effectively used to guide the editing model toward targeted improvements, as the optimization only uses the resulting reward score, and the reward model is non-differentiable. It would be interesting to discuss about methods to better utilize the fine-grained evaluation.

4. The paper claims their method as “better alignment with human preferences,” but there’s no user study or human evaluation confirming the perceptual improvements. Although the reward model is evaluated by prediction accuracy of human preference, the resulting image editing model is evaluated by GPT 4.1.

[1] Inference-Time Scaling for Generalist Reward Modeling. Liu et, al.

**Questions:**

In line 46, the authors claim the reward model leverages test-time computation to improve fidelity. It is unclear to me what this specifically means — does it refer to the CoT reasoning process?

---

### Official Review · Reviewer_WYTj · 2025-11-01

**Soundness:** 2
**Presentation:** 3
**Contribution:** 2
**Rating:** 4
**Confidence:** 4

**Summary:**

This paper proposes Edit-R1, a framework to apply RLHF to image editing using a Chain-of-Thought (CoT) Reasoning Reward Model (RRM). The RRM decomposes instructions into principles and evaluates edits via CoT. The authors introduce Group Contrastive Preference Optimization (GCPO) to train the RRM and subsequently use GRPO to optimize editing models. While the application of RLHF to image editing is a relevant direction, the paper's central claim of novelty—the GCPO algorithm—is highly questionable and not empirically substantiated.

**Strengths:**

- The paper addresses a relevant and under-explored problem: the lack of robust reward models for image editing.
- The idea of using a principle-based, CoT reward model for providing fine-grained feedback is intuitive and potentially valuable.
- The experimental setup for evaluating the reward model and downstream editing tasks is comprehensive.

**Weaknesses:**

- Lack of Novelty in GCPO: The primary weakness is the proposed GCPO algorithm. Upon close inspection, its components appear identical to GRPO. The "group" of rollouts for which an advantage is computed is the core mechanism of GRPO. The "cross-group win/loss ratio" is simply one way to construct a reward signal from preference data, which is standard practice. The "intra-group advantage" is precisely the advantage calculation in GRPO (normalizing rewards by the group mean). The paper presents no clear algorithmic distinction and fails to justify why this should be considered a new algorithm.
- Missing Critical Ablation Study: The paper's failure to compare GCPO with a GRPO baseline for training the RRM is a major oversight. This is the only experiment that could validate the authors' claims. Without it, the observed improvements from the "GCPO" phase could be entirely attributed to the application of GRPO itself, making the contribution of GCPO zero.
- Overstated Claims: The paper claims GCPO is a "novel method" and "significantly improves the quality of the reward model’s inference process." Given the lack of differentiation from GRPO and the absence of comparative evidence, these claims are unsubstantiated and misleading.

**Questions:**

See weakness.

---

### Official Review · Reviewer_XV8w · 2025-11-01

**Soundness:** 3
**Presentation:** 2
**Contribution:** 3
**Rating:** 4
**Confidence:** 3

**Summary:**

The authors propose to use reinforcement learning from human feedback (RLHF) for image editing. Arguing that the key bottlenecks are reliable reward models + the inevitability of its non-differentiability, the authors propose a two step process. First, using Imgedit benchmark, they create a dataset consisting of (i) reference image, (ii) edit instruction, (iii) edited image and (iv) principle set which decompose the edit instruction into a concise set of verifiable principles. This is used for supervised fine-tuning (SFT). Next, to have a refined reward model more attuned to human preferences, they gather a preference data using human annotations. Their final model gets trained using this additional step using a newly proposed reinforcement learning algorithm - Group Contrastive Preference
Optimization. The authors report quantitative results for their reward model evaluation and its usage in making image editing models better.

**Strengths:**

- The idea of breaking down the edit instruction into verifiable principles is a nice idea to make the evaluation of the image editing operation more fine-grained/interpretable, and, consequently, to potentially give better, clear signals to the image editing model to improve.

- Except a few crucial sections, the paper is well written, with all the different stages explained well in text and in figures.

- The application of chain of thought enabled RLHF for the purpose of image editing has been shown for the first time (to the best of my knowledge).

- They report two kinds of (mostly quantitative) evaluations - (i) reward model evaluation measuring whether their reward model prefers image edits preferred by humans, and (ii) image editing model evaluation, measuring how much better their reward model makes existing image editing methods. Improvements are shown in both of these cases, including over a state-of-the-art proprietary reward model.

**Weaknesses:**

- For an image editing paper, there is a surprising lack of qualitative results. In fact, the main paper has no qualitative result showing how the proposed reward model ultimately leads to making image editing models better. The only small section in the main paper is in lines 473-477, linking to a figure in the appendix (Figure 8). And while the few images shown there, along with those in Figure 7, do show cases where the updated model does a better job, I do not think they are sufficiently convincing. For example, I would say that in Figure 7, I can see 6 cases where there is no clear winner. And this is among the few images that the authors have chosen to include in the appendix.

- More qualitative results are needed also because the overall method has a lot of components (more on this point later). There are two phases, and without a qualitative ablation study of those components, claims like those made in lines 209-212 cannot be substantiated.

- While this point could be my misunderstanding, it seems to me that the authors have not explained how the Supervised Fine-Tuning (SFT) dataset, described in Section 3.1.1, will actually be used if one was to only use the first phase. For example, in Table 3, how would a baseline like 'RL w. SFT-RRM (7B)' be obtained?

- There are some questionable mistakes that the authors have made in bolding the results in Table 3. (i) in column corresponding to Category SC - 10 and 'Qwen-Edit Family', the baseline has a higher score than the proposed method but the lower number is bolded. (ii) another case is the column to the right to the previous one, where, again, the baseline model has a higher score, but no score has been bolded. While I can understand that this could have been a mistake, it sends a wrong signal when both the mistakes can be seen to wrongly show that the baseline is not better than the proposed method.

- While the authors are trying to use a new breed of technique (RLHF) for image editing, the resulting method is quite complicated with many intermediate steps. In fact, there needs to be two types of datasets that need to be collected - (i) 200K samples from the Imgedit benchmark for phase 1, and (ii) human preference data for phase 2. The authors should present some kind of computational analysis - how much time it took to train the whole reward model + train downstream image editing models, and also how much time it takes during testing phase to edit an image, compared to baselines. The overall complicated nature of the proposed method is also relevant because we don't get to see much of what qualitative benefits the method brings.

**Questions:**

- Line 290 - "The editing model, acting as the policy πθ(·,c) for each sampling step." -- this seems like an abrupt end to a sentence.

- Line 186: “together with the reference input image x_edit”  -> should be x_ref

- Line 413/424: It should be Figure 4.

---

### Note · Authors · 2025-11-14

I have read and agree with the venue's withdrawal policy on behalf of myself and my co-authors.